# Are boarding secondary schools suitable for students with asthma? An asynchronous online focus group discussion among members of an asthma awareness group

**Kosisochi Chinwendu Amorha**[1,2,3]*, **Kossy Maryann Ochie**[3,4], **Stephen Chukwuma Ogbodo**[3,5], **Olisaemeka Zikora Akunne**[3,6], **Ogechi Christiana Obi**[3,7], **Nwamaka Theresa Ene**[3,8], **Chukwudi Richard Ifeanyi**[3,9], **Jonathan Ikokwu**[3,10], **Chibuike Victor Eze**[1,3], **Emmanuella Tochukwu Ogbonna**[1,3], **Marydith Ifeoma Chukwu**[3,11], **Chinedu Collins Okafor**[3,12], **Chiamaka Ruth Echeta**[1,3], **Somtochi Prosper Nwani**[3,13], **Christabel Ogechukwu Okoye**[1,3], **Vanessa Chinweike Okonkwo**[1,3], **Chisom Jennifer Eneje**[1,3], **Gerald Obinna Ozota**[1,3]

1 Department of Clinical Pharmacy and Pharmacy Management, Faculty of Pharmaceutical Sciences, University of Nigeria Nsukka, Enugu, Enugu State, Nigeria, 2 Department of Clinical Pharmacy and Pharmacy Practice, Faculty of Pharmaceutical Sciences, Bingham University, Karu, Nasarawa State, Nigeria, 3 Asthma Awareness and Care Group (AACG), Nsukka, Enugu State, Nigeria, 4 Department of Clinical Pharmacy and Pharmacy Management, Nnamdi Azikiwe University, Awka, Anambra State, Nigeria, 5 Pharmacy Department, Enugu State University Teaching Hospital, Enugu, Nigeria, 6 ASK Medical and Diagnostic Center, Federal Capital Territory, Abuja, Nigeria, 7 Federal Medical Centre, Umuahia, Abia State, Nigeria, 8 Kareda Pharmacy, Asaba, Delta State, Nigeria, 9 National Institute for Pharmaceutical Research and Development, Abuja, Nigeria, 10 Pharmacy Council of Nigeria, Lagos Zonal Office, Yaba, Lagos State, Nigeria, 11 Federal Medical Center, Lokoja, Kogi State, Nigeria, 12 Lyn Edge Pharmaceuticals Limited, Lagos, Nigeria, 13 Medplus Pharmacy Nigeria Limited, Lagos, Nigeria

* kosisochi.amorha@unn.edu.ng

**Data Availability Statement:** All relevant data are within the paper.

## Abstract

### Introduction

Children in boarding schools spend most of their time without their parents or caregivers, causing concerns about the suitability of such schools for children with asthma. This study assessed individuals' opinions regarding the suitability of boarding secondary schools for children with asthma.

### Methods

A qualitative design was adopted for this study using a focus group discussion held on a social media platform (WhatsApp®) of the Asthma Awareness and Care Group (AACG), The group comprised 150 registered members. The study was guided by a structured protocol and based on a vignette comprising three questions. Data were analysed via thematic analysis using framework principles.

**Funding:** The authors received no specific funding for this work.

**Competing interests:** The authors have declared that no competing interests exist.

## Results

Out of the 150 eligible members, there were responses from only 19 participants. Majority of the respondents were aged $\leq$ 30 years (n = 17, 89.5%). The three main themes generated from the thematic analysis include the appropriateness of boarding schools for children with asthma; facilities necessary for boarding schools to cater to children with asthma; and outright rejection of children with asthma by boarding schools. Respondents conceptualised the appropriateness of boarding schools for students with asthma in six distinct sub-themes: asthma severity and extent of control, child's self-efficacy and assertiveness, child equipped with tools (knowledge, inhalers, and asthma control diary), school awareness, facilities, and active support, availability of a guardian, and the knowledge and perception of teachers and schoolmates about asthma. The sub-themes associated with the themes were presented, alongside exemplar quotes from respondents. The majority of the respondents (61.5%) were in support of allowing children with asthma attend boarding schools but with some caveats such as without liability to the school, if facilities are unavailable.

## Conclusion

Children's age, autonomy, asthma management status, and the school's readiness were identified as important considerations for the safe attendance of children with asthma at boarding schools.

## Introduction

Childhood asthma, recognised as the most common chronic respiratory disease among children, poses the highest disability burden in this population and accounts for most cases of school absenteeism [1, 2]. Asthma is a multifactorial disease caused by a combination of genetic, host, and environmental factors [3, 4]. The global prevalence of asthma has increased significantly in the last forty years, with the World Health Organization (WHO) estimating that 300 million people currently live with asthma, and this is projected to reach 400 million by 2025 [1]. In terms of mortality, 250,000 people die from asthma annually, with a reported mortality rate of up to 0.7 per 1000 among children [1, 5].

Nigeria has one of the highest burdens of childhood asthma in Africa, with a rapidly increasing prevalence from 10.7% in 1999 to about 20% in 2014 [6]. Nigeria currently follows the Universal Basic Education (9-3-4 system of education), a transition from the 6-3-3-4 system [7]. The 9-3-4 system entails nine years of basic education, three years of senior secondary education, and four years of tertiary education. The curriculum for the 9-3-4 system of education was designed to meet the Millennium Development Goals (MDGs) by 2020 [7]. Students can be in boarding schools from the seventh to the ninth year of basic education and during the three years of senior secondary education. There is an option of having children going from home (day students) or living and studying in the school during the school term (boarders) [8].

In Nigeria, five of every 100 adolescents in Nigeria who are school children, especially those in secondary schools, are in boarding schools [9]. Many parents or caregivers lean towards the option of boarding schools as it helps children build independence, limits access to distracting

technology, and provides an established learning community where they can easily form study groups [10].

Since basic education is a legal right in Nigeria, as with most other countries, the management of asthma among children within the school system has become a central issue [11]. The Global Initiative for Asthma (GINA) estimated that 14% of school-aged children have had at least one episode of wheezing in the past 12 months [12]. For children attending "day" schools, their parents or caregivers have ample opportunity to monitor and manage their asthma, as they return home, daily, after school. This is less feasible for boarding school children who enjoy significantly reduced parental oversight. Hence, boarding school children with asthma often have to take up greater responsibility for managing their triggers, symptoms, and attacks alone or with the help of the school management. As such, many parents/guardians of children with asthma may be skeptical about sending their children to boarding schools.

It has been documented that children with asthma face significant challenges in school which can affect their academic performance and quality of life [11]. Acute asthma attacks in school children account for 2–10% of emergency hospital visits in Nigeria, largely because most schools lack appropriate facilities and health services to manage asthma episodes [13, 14]. Students with asthma may also experience stigmatisation through exclusion from school programmes like sports and other physical activities [15, 16]. Although studies have shown that incorporating asthma-related topics in school curricula will improve students' and teachers' knowledge of asthma and improve the experience of children with asthma, parents and caregivers are still faced with the dilemma of the suitability of boarding schools for their children with asthma [17, 18].

Thus, this study was designed to evaluate individual perceptions and opinions regarding the appropriateness of boarding schools for students with asthma, particularly in Nigeria. It further aimed to elicit perspectives about the facilities, appropriate training, and conditions that would enable the effective management of students with asthma in boarding schools.

## Methods

### Study design

This was a qualitative study involving a focus group discussion held on the social media platform (WhatsApp®) of the Asthma Awareness and Care Group (AACG). The discourse, which was held on 22 March 2023, was guided by a structured protocol. It was based on a case study comprising three questions. The vignette was posted in the Group, without previous publicity. Members of the Group were informed that only comments received within a specified time frame would be utilised for the study (14:00–21:00 hours). A 7-hour time frame was fixed, considering that the discussion was impromptu. This asynchronous online focus group (AOFG) research design using a specified timeframe has been recommended due to the flexibility it affords respondents to participate in qualitative research [19, 20]. The discourse was moderated by the Founder of the Group. AACG is a team of asthma enthusiasts who aim to improve asthma awareness and care, in communities [15].

### Ethical approval

The study did not require ethical approval. It was a focus group discussion conducted online, in a closed group on WhatsApp®. The group is restricted to members of the Asthma Awareness and Care Group (AACG). Participation of members of the Group was voluntary. Members who participated were informed that their comments would be extracted for research purposes. They were also informed that the authors would be members of AACG who could identify individual participants during or after data collection.

## Eligibility criteria

Eligibility criteria comprised members of the Asthma Awareness and Care Group (AACG) who were registered on the AACG WhatsApp® platform. As of the time the study was conducted, there were 150 registered members on the AACG WhatsApp® platform. Members of the Group were majorly Pharmacists (n = 142, 94.7%), with a large proportion having not more than 10 years of experience. The Pharmacists practiced in community, hospital, academia, industry, and administrative pharmacy settings. Other members of the Group included: five undergraduate pharmacy students, one Public Relations Professional, one Economist, and one Mechanical Engineer. Only comments received within the 7-hour time frame were included.

## Sample size and selection

The sample size was not calculated. All registered members of the AACG WhatsApp® platform were eligible for participation. A convenience sampling technique, within a specified timeline, was employed. Comments were received from 19 participants who responded within the allotted timeframe for data collection.

## Data collection

This vignette was posted on the AACG WhatsApp® platform:

AB is an 11-year-old known patient with asthma.

As the closest healthcare provider to her family:

1. *Would you advise AB to go to a Boarding School? Kindly justify your position.*

2. *What facilities would you advise Boarding Schools to have, to cater to students with asthma?*

3. *Should Boarding Schools simply reject patients with asthma i.e., better safe than sorry?*

The comments from the participants provided data for the study. Demographic data were collected for each respondent.

## Data analysis

After the group discussion, a thematic analysis was conducted using framework analysis principles and generally adhering to the five stages of familiarization: identifying a thematic framework, indexing, charting, mapping, and interpretation.

For the identification of a thematic framework, nine researchers independently acquainted themselves with the transcripts, acquiring a general feel for the data and noting emergent ideas on the margins. The researchers identified key themes in the transcript. This was similar to a study conducted on the challenges and expectations of the Mental Capacity Act of 2005 where an in-depth qualitative methodology was adopted. Three a priori questions formed the thematic framework, based on the codes and sub-themes that were subsequently identified [21, 22]. The thematic framework included three central themes: 1) appropriateness of boarding schools for children with asthma; 2) facilities necessary for boarding schools to cater for children with asthma; and 3) outright rejection of children with asthma by boarding schools.

For the "indexing" stage, the nine researchers were assigned different parts of the transcripts, which they iteratively read to identify ideas and assign codes to portions of text. This generated a coding framework. This process was repeated until no new codes were generated. Discrepancies between the nine researchers' codes were resolved through discussion and a unified coding framework was obtained [22]. These codes were initially identified under the

three broad themes in the thematic framework, after which similar codes were clustered to form sub-themes.

In the "charting" step, these themes, subthemes, and codes were presented in a chart and discussed at team meetings, enabling the re-ordering of the subthemes as necessary. This made it possible to extract relevant details from the data, including elements that may have been missed if a purely a priori approach was employed [23].

In the "mapping and interpretation" stages, the themes and subthemes were tabulated alongside exemplar quotes from participants. This process produced a 'tree structure' featuring interconnected themes, subthemes, and supporting quotes. Finally, these identified ideas were interpreted to answer the research questions and identify areas for public health action.

## Results

Nineteen respondents participated in the focus group discussion, comprising 11 females and eight males. Majority of the respondents were aged ≤ 30 years (n = 17, 89.5%), Table 1.

The thematic analysis produced three themes: the appropriateness of boarding schools for children with asthma (Table 2); the facilities necessary for boarding schools to cater to children with asthma (Table 3); and the outright rejection of children with asthma by boarding schools (Table 4).

Respondents conceptualised the appropriateness of boarding schools for students with asthma according to six distinct sub-themes: asthma severity and extent of control, child's self-efficacy and assertiveness, child equipped with tools (knowledge, inhalers, and asthma control diary), school awareness, facilities, and active support, availability of a guardian, and the knowledge and perception of teachers and schoolmates about asthma.

**Table 1. Socio-demographic characteristics of the participants, N = 19.**

| Participant Number | Gender | Age category (years) |
|---|---|---|
| P1 | F | 26–30 |
| P2 | M | 26–30 |
| P3 | M | 26–30 |
| P4 | F | 18–25 |
| P5 | M | 26–30 |
| P6 | M | 26–30 |
| P7 | F | 18–25 |
| P8 | F | 26–30 |
| P9 | M | 26–30 |
| P10 | F | 26–30 |
| P11 | F | 18–25 |
| P12 | F | 18–25 |
| P13 | F | 26–30 |
| P14 | M | > 30 |
| P15 | F | 26–30 |
| P16 | F | 26–30 |
| P17 | M | 26–30 |
| P18 | M | > 30 |
| P19 | F | 18–25 |

M = Male; F = Female

All respondents were Pharmacists, except P11 and P12 who were in undergraduate pharmacy school

**Table 2. Appropriateness of boarding schools for students with asthma.**

| Theme | Subthemes | Quotes |
|---|---|---|
| Appropriateness of boarding schools for children with asthma | 1. Asthma severity and extent of control | (a) "I would advise AB to go to a boarding school if her asthma is well-controlled. In general, I would judge the severity of her condition". (P2) |
| | | (b) "It would depend on her level of asthma control. That she is a known patient with asthma does not necessarily imply that she has poorly-controlled asthma. It is possible that she might have well-controlled or even perfectly-controlled asthma". (P18) |
| | | (c) "Making sure AB stays safe with asthma at a boarding school can feel a bit daunting. However, I would advise AB to go to a boarding school if her asthma is well-controlled". (P17) |
| | | (d) 'Since AB is a known asthma patient, I want to assume that her parents have placed her on the right medications. If she is properly managed, that means if her Asthma Control test is 20 and above, she can go to the boarding school". (P13) |
| | 2. Child's self-efficacy and assertiveness | (a) "Has AB been taught how to stand up for herself with regards to her health status, respectfully but firmly? For example, she can say "Sorry I can't participate in sweeping because it is too dusty and I'm allergic to dust". (P1) |
| | | (b) "AB's ability to take care of herself without supervision should be considered. AB's sense of responsibility and knowledge about her health condition should be considered too. Checking all of these boxes are huge expectations from an 11-year-old, even in the best-case scenarios". (P5) |
| Appropriateness of boarding schools for children with asthma | 3. Child equipped with tools: knowledge, inhalers, asthma diary | (a) "She can go but her parents must equip her with her inhaler, asthma control diary and other things she needs to be successful while in school". (P1) |
| | | (b) "AB can be advised to go to a boarding school if there's proper knowledge about her asthma triggers, extent of exacerbation, and proper management plan". (P6) |
| | | (c) "The child should become conversant with managing her condition and avoiding her triggers and not always needing external help because she won't remain a child". (P12) |
| | | (d) "Irrespective of the level of her asthma control, it is important to find out if she knows her asthma triggers and her level of asthma self-efficacy. At 11, and with a history of asthma, her knowledge of asthma self-management will need to be brought to the fore in a boarding school". (P18) |
| | 4. School awareness, facilities and active support | (a) 'I will also go to inform the school authorities of her condition and her dos and don'ts". (P2) |
| | | (b) "Yes, I would advise AB to go to a boarding school because the one I went to, made sure we submitted our blood test results and other health challenges before we were admitted. The school also had the Red Cross Society in charge of administering First Aid in case of any Emergency". (P4) |
| | | (c) "Boarding schools can be safe and appropriate environments for students with asthma if they have adequate facilities and support systems in place to manage their condition." (P3) |
| Appropriateness of boarding schools for children with asthma | School awareness, facilities and active support | (d) "Though I'd say it's okay to take the child to a boarding school if the condition is not so severe, as long as the school has what it takes to manage asthmatic situations when it arises." (P12) |
| | 5. Availability of a guardian | (a) "But in a case where she doesn't have a proper knowledge about the condition, the parents should hand her over to a guardian or tell the school authorities. I can vividly remember, then in school, asthmatic students were given some kind of preferential treatment, like bathing in hot water, not allowed to run or even do some manual labours and so on". (P10) |
| | | (b) "If being in a boarding school is necessary, then it is okay for her to go. However, her parents would have to assign a guardian to check with her regularly". (P15) |
| | 6. Knowledge and perception of teachers and schoolmates about asthma | (a) "We also knew those who were Asthmatic, and in case of an emergency, you knew who to ask for an inhaler during crises" (P4) |
| | | (b) "Students should also be enlightened on what to do when someone has a crisis". (P10) |

*(Continued)*

**Table 2.** (Continued)

| Theme | Subthemes | Quotes |
|---|---|---|
| Appropriateness of boarding schools for children with asthma | 7. Boarding school not advised | (a) "If there's any chance at all that AB can go to school from home, then AB shouldn't consider a boarding school at such a young age". (P5) |
| | | (b) "AB is young. Triggers for asthma attacks are peculiar to individuals. It's possible that most of AB's triggers have not even been known, the change of environment, coupled with the emotional unrest that comes with moving into a boarding school could all be triggers on their own for such a young patient. It's going to be a whole new world for AB in a boarding school, with exposure to a lot of foreign factors/allergens. An 11 y/o wouldn't be able to properly manage all of it". (P8) |
| | | (c) "I would NOT advise AB to go to a boarding school because AB is still young and may not know how to take care of herself without proper supervision. Also, AB's triggers might not be known, completely". (P11) |
| | | (d) "I would not advise an 11-year-old patient with asthma to go to a boarding school because; I feel she's still very tender and may not be able to deal with the condition, alone, in a boarding school. The living conditions in most boarding schools may not be suitable for her health. Being that she may not have developed adequate understanding and awareness of his condition (at her age), she may not know how to identify her triggers and avoid them. Even if she does know the triggers, what if she can't avoid them, given that it's a public home?" (P8) |
| Appropriateness of boarding schools for children with asthma | Boarding school not advised | (e) "I would not advise AB to go to a boarding school. That is a young age for a child with asthma to be in boarding school whether the asthma is well-controlled or not. Triggers can happen anytime and I believe at that age the child might not be fully aware of all the them. Even adults get triggered at any point, much less an 11-year-old trying to adjust to a new environment." (P16) |
| | | (f) "My cousin was in a boarding school but she recently had to leave because her asthma was not properly managed. As long as there are alternative day-schools that are good, I don't see the reason why the child should be put at risk at such a young age". (P11) |
| | | (g) "Most asthma-related deaths occur in low- and lower-middle-income countries, where under-diagnosis and under-treatment is a challenge- (WHO) and I believe that the best way to prevent these complications is to avoid asthma triggers and this is often difficult in boarding schools where there are several shared facilities and spaces than in day school. There is a higher chance of disease spreading such as flu which could trigger attacks". (P19) |
| | | (h) "Part of the reason why parents send their kids to boarding schools is to help them live independently but an 11-year-old with asthma away from their parents might only lead to a lot of micromanaging which in this case defeats the aim. The child is better off in a day school where they can be monitored daily at home which by research is recommended". (P19) |

The subthemes associated with each of these themes are presented, alongside exemplar quotes from respondents.

## Discussion

The findings of this study address the issue of appropriateness of boarding schools for children with asthma. The respondents were stakeholders involved in asthma care and knowledgeable about the resources necessary for adequate management of asthma.

### Theme 1: Appropriateness of boarding schools for children with asthma

The question of whether or not boarding schools are appropriate for students with asthma is a complex issue that requires careful consideration of several factors. The opinions expressed by the respondents indicate that there is no clear consensus on the matter and that the decision should be based on a case-by-case evaluation.

**Table 3. Facilities necessary for boarding schools to cater for students with asthma.**

| Theme | Subtheme | Quotes |
|---|---|---|
| Facilities necessary for boarding schools to cater to students with asthma | 1. Adequate and functional on-site medical facilities with round the clock availability of health professionals e.g., nurses | (a) "They must have a functional sick bay with a trained nurse on standby and the sick bay must be stocked with inhalers and even nebulizers in cases of emergency". (P1) |
| | | (b) "It will also be necessary to know if the school has health professionals in their sick-bay or clinic, 24/7, who have the knowledge and expertise (as well as medications and devices), to act as responders if there is severe asthma exacerbation". (P18) |
| | | (c) "A sick bay with a trained nurse with empathy because ours was trained but lacked empathy". (P4) |
| | | (d) "Boarding schools should have trained health personnel with updated knowledge of asthma management. They should have a clinic equipped with asthma relievers to help manage any asthma episodes and emergencies". (P8) |
| | | (e) "A boarding school should have emergency asthma kits such as nebulizers, inhalers and oxygen devices. Also, they should have an ambulance for carrying patients when referred". (P2) |
| | | (f) "They should have a medical staff with proper knowledge and orientation about asthma intervention and management, an equipped drug store with essential asthma medications and inhalers, and a sick bay". (P6) |
| Facilities necessary for boarding schools to cater to students with asthma | 2. Effective referral system for severe emergencies. | (a) "The school must be affiliated with a nearby hospital where they can rush the students to, in cases of emergency like status asthmaticus". (P1) |
| | | (b) "There should be a retainership agreement with a standard hospital or a Pulmonologist to manage serious cases before the physical arrival of parents/guardians". (P14) |
| | | (c) "The school should have a hospital subsidiary to facilitate referral systems in severe emergencies". (P17) |
| | 3. Measures to minimize exposure to environmental triggers: dust, pollen, cold. | (a) "The school should take note of the students with asthma and ensure as much as possible that they are not exposed to chores that could trigger an attack". (P1) |
| | | (b) "Boarding schools should have facilities such as clean and well-ventilated dormitories, classrooms, and common areas to reduce exposure to triggers such as dust, molds, and pet dander". (P3) |
| | | (c) "A boarding school can be safe and able to cater for asthmatic students if it has less exposure to most common allergens including dust and pets, clear routines that limit allergen induction like controlled fumigation practice especially during holidays when students are away". (P6) |
| | 4. Collaboration and communication between parents, school, guardian and the child | (a) "It is also important for schools to have clear communication with parents or guardians about their child's asthma care plan, medication requirements, and emergency contact information". (P3) |
| Facilities necessary for boarding schools to cater to students with asthma | | (b) "It's very paramount that the school and the parents work as partners; ensuring proper communication, provision of medications and necessary resources to keep children with asthma, safe in school". (P19) |
| | 5. Training for staff and students to recognise asthma signs and help | (a) "They should have trained staff who are familiar with asthma management, including recognizing symptoms, administering medication, and responding to emergencies". (P3) |
| | | (b) "The school should have a Red Cross Society which is an organization of students like you who are continuously trained on various crisis situations that may arise and how to deal with it". (P4) |
| | | (c) "Have teachers and management who are not just aware but have positive perceptions about asthma (You wouldn't want to send a child to a school where the teachers or management believes that asthma is infectious or that asthmatic attacks are solved by persons surrounding the asthmatics and pouring him/her water)". (P14) |
| | | (d) "Staff of the school should be First Aid-trained and aware of what to do in an asthma attack and how to use a reliever inhaler. Further training can be offered to individual staff that are supporting children to use their inhalers". (P17) |

**Table 4. Outright rejection of children with asthma by boarding schools.**

| Theme | Subthemes | Quotes |
|---|---|---|
| Outright rejection of children with asthma by boarding schools | 1. Never reject–to avoid discrimination and stigmatization | (a) "No, boarding schools should not simply reject students with asthma. Asthma is a common chronic condition, and with proper management and support, students with asthma can thrive academically and socially. Rejecting students with asthma is discriminatory and can limit their opportunities for personal and academic growth. Instead, boarding schools should work to create an inclusive and supportive environment for all students, including those with asthma. This can include providing appropriate facilities and resources, as well as promoting education and awareness about asthma among students, staff, and faculty" (P3) |
| | | (b) "No! It could be misconstrued as stigmatization". (P18) |
| | 2. Reject if unprepared—better safe than sorry | (a) "Only if they are not equipped. I think it's best for everyone if ill-equipped boarding schools reject known asthmatics. I can understand why other colleagues feel differently, as "rejection" has strong connotations of discrimination and unequal opportunity. But, *better safe than sorry*"! (P9) |
| | | (b) "Refusing a patient because they have asthma can be termed discriminatory. However, if they are not equipped, accepting such a child is simply criminal". (P14) |
| | | (c) "A boarding school should not reject a student with asthma. They can do so if they know fully well, they are not properly equipped and can't handle the situation". (P10) |
| | 3. Accept even if unprepared but without liability to the school | (a) "A boarding should not outrightly reject a student as this can be a form of stigma, they should focus on been equipped but if all these are not available, it is best they are clear on the extent of service they can offer for such emergency to the parent or guardian and the decision left on them to make if they would want to put their child or ward or not". (P2) |
| Outright rejection of children with asthma by boarding schools | Accept even if unprepared but without liability to the school | (b) "They should advise parents to opt for day school. However, if parents insist school should not be held liable". (P19) |
| | 4. Accept if prepared | (a) "Where boarding schools should not outrightly reject an asthma student, they must ensure that they are fully equipped to take care of those children". (P1) |
| | | (b) "Boarding schools should not reject patients as long as they have a good facility to take care of them to avoid asthma being like a social stigma". (P11) |
| | | (c) "No, boarding schools should not reject all asthma patients. What they can do is to make sure that the patient clearly informs the school and staff that needs to know and that the patient is properly managed. Make sure that the patient is compliant with medications and safety practices." (P13) |

The majority of the participants (61.5%) consented that children with asthma can be allowed to attend boarding schools with strict conditions. For instance, asthma must be well-controlled based on the Asthma Control Test™ (ACT™), the children must be well-equipped by their parents or caregivers with skills for asthma self-management such as proper inhaler and asthma diary use, they must be assertive about their rights, and have good support systems in form of teachers and schoolmates who are trained and have a positive perception of asthma. A study opined that parents should inform teachers about their children's written asthma action plans to guide teachers during emergencies, rather than waiting for parents to pick up their children from school while taking no life-saving actions [24]. Another study documented that school-based educational interventions led to improvement in teachers' knowledge, self-management, and health outcomes of asthma students [25]. All these are to abate the poorer quality of life, frequent emergency department visits, hospitalizations, and school absenteeism that emanates from uncontrolled asthma [26, 27].

Some participants believed that boarding schools are not appropriate for students with asthma, regardless of the asthma severity and control. They cited concerns about the change of environment, exposure to allergens (including unknown ones), emotional unrest, lack of supervision, and the living conditions in most boarding schools in Nigeria. This opinion is similar to several studies that have reported that children with asthma are at increased risk of being bullied and victimised in school [28]. Another survey conducted in Nigerian primary

and secondary schools leaves much to be desired as the majority of public schools in rural and poor urban slums have substandard environments that are suboptimal for not just students with asthma but non-asthmatics as well [29, 30].

## Theme 2: Facilities necessary for boarding schools to cater for students with asthma

From the responses provided, it is evident that there is a consensus on the facilities required to manage students with asthma in boarding schools. This indicates that there are measures that must be put in place by boarding schools to cater to the needs of students with asthma. These facilities include adequate and functional medical facilities with health professionals such as nurses, emergency asthma kits, and an ambulance to transport students to the hospital when necessary. Unfortunately, this seems not to be the case as an assessment of school health services in Nigeria revealed poor and sub-standard health facilities in most public and even private schools [14]. A study in Lagos State, Nigeria, revealed that only 16% out of 54 schools surveyed had a school clinic; only 7.4% had a school health worker, and none of the schools had facilities for asthma emergency care [31].

Findings from this study also indicate that effective referral systems for severe emergencies are necessary, including retaining an agreement with a standard hospital or a pulmonologist. Additionally, the school must minimise exposure to environmental triggers like dust, pollen, and cold. It is also essential that the school and parents/guardians work together to provide proper communication, medication, and necessary resources to keep children with asthma safe in school. Training and re-training of staff and students to recognize asthma symptoms and provide help is crucial.

## Theme 3: Outright rejection of children with asthma by ill-equipped boarding schools

Regarding the outright rejection of children with asthma by boarding schools, most respondents agreed that boarding schools should not simply reject students with asthma. Instead, boarding schools should provide the necessary facilities and resources to create an inclusive and supportive environment for all students, including those with asthma. Asthma is a common chronic condition, and with proper management and support, students with asthma can thrive academically and socially. Therefore, rejecting students with asthma is discriminatory and can limit their opportunities for personal and academic growth.

However, some respondents further noted that boarding schools should reject known asthmatics if they are not equipped to take care of them. This is to avoid any negative consequences that may arise from accepting students without adequate facilities to manage them. If a boarding school is not equipped to handle students with asthma, they should inform the parents or guardians and advise them to opt for day school. Some participants thought that even schools that do not have all it takes to manage such conditions in place could also accept such students, on the condition that they explain their shortcomings to the parents or guardians of these children. This could prevent their actions from being misconstrued as the stigmatisation of children with asthma and might ensure parents/guardians are held responsible for whatever happens to the child if they eventually decide to enrol.

This study is an attempt to evoke a thought process on whether children with asthma should be in boarding schools. It suggests that the suitability of boarding schools for children with asthma is dependent on many factors. Boarding schools should create an inclusive and supportive environment for students with asthma.

Boarding school administrators should ensure that they have adequate facilities to manage patients with asthma. This includes a functional sick bay and health professionals who would be available whenever needed. The sick bay should have medicines for the management of acute and chronic asthma, as well as asthma devices such as peak flow meters and nebulisers.

The boarding school should have links with registered hospitals that can handle referrals. There should be collaborative efforts amongst the teachers, students, parents, and school administrators to ensure students with asthma have a good quality of life while at school.

Asthma education programmes should be conducted for all active players (students, parents, teachers, and school administrators). There should be training and re-training for staff and active players who can serve as first responders if there is an asthma attack.

The boarding school should have asthma-friendly environments. All possible asthma triggers should be eliminated.

Policymakers and those who approve schools to take boarding students should ensure that the facilities and environments are suitable for patients with asthma. Lists of items that need to be put in place should be provided to school administrators. Policymakers should form a think tank using this study as a baseline to provide a policy framework that will guide the educational sector to make important changes to the boarding schools that would host students with asthma.

Future studies should consider quantitative assessments. Questionnaires could be administered to a larger sample size to quantify the prevalence of specific opinions and perceptions related to boarding schools for children with asthma. Longitudinal studies could also be conducted to provide valuable insights into the dynamics of opinions regarding boarding schools for children with asthma and allow for a deeper understanding of the factors influencing these perspectives. Mixed-method analysis could be done to better understand the experiences and health outcomes of children with asthma attending boarding schools in comparison to those attending day schools.

Building upon the findings of this study, future research could focus on developing and implementing interventions aimed at improving the management and support systems for children with asthma who are in boarding schools. Evaluating the effectiveness of interventions, such as asthma education programmes for teachers and students, improved medical facilities and resources, and strategies to reduce stigma and promote inclusivity could provide evidence-based recommendations for creating asthma-friendly environments in boarding schools.

## Limitations of the study

The study was conducted within the confines of a closed social media group, the Asthma Awareness and Care Group (AACG), which is predominantly made up of healthcare professionals and enthusiasts involved in asthma care. These participants may possess a higher level of knowledge and expertise regarding asthma management than the general population and may not fully represent a general perspective on the subject matter. Furthermore, their expertise and familiarity with the subject matter might have influenced their responses and recommendations. In addition, the study employed convenience sampling without calculating the sample size. As a result, the sample size may not be sufficiently large or diverse to capture the wide range of opinions and experiences related to boarding schools and asthma management in Nigeria. Future studies should consider a more diverse range of participants.

## Conclusion

This study, in an attempt to address the issue of the appropriateness of boarding schools for children with asthma, has identified children's age, autonomy, asthma management status,

and the school's readiness, as important considerations for the safe attendance of children with asthma at boarding schools. With childhood asthma being common, we therefore recommend, that schools put measures in place to ensure that students with asthma can lead active lives. They should be able to learn in a safe environment without fear of stigmatisation or lack of facilities/medications/expertise/referral systems, in emergencies.

## Acknowledgments

The authors appreciate all members of the Asthma Awareness and Care Group (AACG) for their participation and responses which were the basis for the study. AACG also appreciates the Respiratory Pharmacists of Nigeria (RPN) which is a specialty group under the Clinical Pharmacists Association of Nigeria (CPAN). The case study used in this research was coined by Pharmacist Billy Shoaga of RPN/CPAN.

## Author Contributions

**Conceptualization:** Kosisochi Chinwendu Amorha.

**Formal analysis:** Kosisochi Chinwendu Amorha, Kossy Maryann Ochie, Stephen Chukwuma Ogbodo, Chiamaka Ruth Echeta.

**Methodology:** Kosisochi Chinwendu Amorha, Chukwudi Richard Ifeanyi, Marydith Ifeoma Chukwu.

**Supervision:** Kosisochi Chinwendu Amorha.

**Writing – original draft:** Kosisochi Chinwendu Amorha, Kossy Maryann Ochie, Stephen Chukwuma Ogbodo, Olisaemeka Zikora Akunne, Ogechi Christiana Obi, Nwamaka Theresa Ene, Chukwudi Richard Ifeanyi, Jonathan Ikokwu, Chibuike Victor Eze, Emmanuella Tochukwu Ogbonna, Marydith Ifeoma Chukwu, Chinedu Collins Okafor, Chiamaka Ruth Echeta, Somtochi Prosper Nwani, Christabel Ogechukwu Okoye, Vanessa Chinweike Okonkwo, Chisom Jennifer Eneje, Gerald Obinna Ozota.

**Writing – review & editing:** Kosisochi Chinwendu Amorha, Kossy Maryann Ochie, Stephen Chukwuma Ogbodo.

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
