## [Decision Letter · Decision Letter 0]

6 Sep 2023

PONE-D-23-13651Are boarding secondary schools suitable for students with asthma? A focus group discussionPLOS ONE

Dear Dr. Amorha,

Thank you for submitting your manuscript to PLOS ONE. After careful consideration, we feel that it has merit but does not fully meet PLOS ONE’s publication criteria as it currently stands. Therefore, we invite you to submit a revised version of the manuscript that addresses the points raised during the review process.

Reviewer 1:Overall, the manuscript addresses an important topic and provides valuable insights. However, there are some areas that require attention and revision.

Major Comments:

1. Sample Size and Participant Selection: The manuscript does not provide a clear rationale for the selection of the 19 respondents from the initial 150 participants. It is important to provide a justification for the final sample size and explain any potential biases or limitations associated with the selection process.

2. Thematic Analysis: While the manuscript mentions that thematic analysis was conducted, it lacks details about the specific steps and analytical framework used. Providing more information on the process of thematic analysis and the specific themes that emerged would enhance the transparency and rigor of the study. Additionally, it would be helpful to include a brief discussion on the validity of the findings.

3. Generalizability: The manuscript should acknowledge the limitations regarding generalizability. The study was conducted within a specific social media group and included participants who are actively involved in asthma care. Therefore, the perspectives presented may not fully represent the broader population. It is recommended to discuss the limitations and suggest future research directions to include a more diverse range of participants.

4. Recommendations and Implications: The manuscript briefly mentions recommendations for boarding schools to create an inclusive and supportive environment for students with asthma. However, these recommendations should be further developed and expanded upon to provide actionable suggestions for policymakers, school administrators, and healthcare professionals. Consider discussing specific strategies for improving asthma management in boarding schools, such as the training of staff, collaboration with healthcare providers, and development of asthma-friendly environments.

Minor Comments:

1. Geographical Context: Consider providing additional contextual information on the Nigerian education system, including the prevalence of boarding schools and their characteristics. This would help readers better understand the specific context in which the study was conducted.

2. Limitations and Future Research: While this study provides valuable insights into the perceptions and opinions regarding boarding schools for children with asthma, it is important to acknowledge its limitations and identify areas for future research.

(a) Quantitative Assessment: This study adopted a qualitative design using focus group discussions. While qualitative methods are valuable in exploring in-depth perspectives and experiences, future research could complement these findings with quantitative assessments. Surveys or questionnaires could be administered to a larger sample size to quantify the prevalence of specific opinions and perceptions related to boarding schools for children with asthma. This would allow for statistical analysis and a more comprehensive understanding of the overall attitudes towards this topic.

(b) Longitudinal Studies: The current study provides a snapshot of participants' opinions at a specific point in time. Conducting longitudinal studies would enable the examination of changes in attitudes and perceptions over time. This would provide valuable insights into the dynamics of opinions regarding boarding schools for children with asthma and allow for a deeper understanding of the factors influencing these perspectives.

(c) Comparative Analysis: To gain a better understanding of the suitability of boarding schools for children with asthma, future research could compare the experiences and outcomes of children with asthma attending boarding schools versus day schools. This comparative analysis would shed light on the potential advantages and disadvantages of each educational setting and help inform policymakers, school administrators, and parents/guardians in making well-informed decisions.

(d) Intervention Studies: Building upon the findings of this study, future research could focus on developing and implementing interventions aimed at improving the management and support systems for children with asthma in boarding schools. Evaluating the effectiveness of interventions, such as asthma education programs for teachers and students, improved medical facilities and resources, and strategies to reduce stigma and promote inclusivity, would provide evidence-based recommendations for creating asthma-friendly environments in boarding schools.Reviewer 2:The manuscript is very rich in vital scientific information that is of immense benefit to public and school health, which has been a neglected tropical issue. However, my major concern is the fact that out of 150 participants only 19 respondents' responses were available and used for discussions and drawing conclusions. The question now is, are these 19 responses sufficient to make the conclusion seen in this manuscript?

Secondly, why was sample size not determined for the study? This would have ensured that more responses and statistically significant population responses for concrete conclusion is gotten.

Thirdly, you stated that majority of respondents were is support of allowing children with asthma go to boarding school. What's your definition of majority? It is better you state the majority in numerical terms.

The use of social media and WhatsApp differently is a mere repetition.

The method(s) used were not properly cited and where modifications exist were not spelt out.

Many statements made in the discussion were not related to corroborate the work of others, it was mere hanging statement.

Some of the statements were also just made without linkages.

If the research was funded, the sponsors need to be acknowledged.

We look forward to receiving your revised manuscript.

Kind regards,

Babatunde Adewale

Academic Editor

PLOS ONE

Journal Requirements:

Reviewers' comments:

Reviewer's Responses to Questions

**Comments to the Author**

1. Is the manuscript technically sound, and do the data support the conclusions?

Reviewer #1: Yes

Reviewer #2: Yes

2. Has the statistical analysis been performed appropriately and rigorously? 

Reviewer #1: Yes

Reviewer #2: Yes

3. Have the authors made all data underlying the findings in their manuscript fully available?

Reviewer #1: Yes

Reviewer #2: Yes

4. Is the manuscript presented in an intelligible fashion and written in standard English?

Reviewer #1: Yes

Reviewer #2: Yes

5. Review Comments to the Author

Reviewer #1: General Comments: This manuscript presents an investigation into the opinions and perceptions regarding the appropriateness of boarding schools for children with asthma, particularly in Nigeria. The study adopts a qualitative design using a focus group discussion on a social media platform (WhatsApp). The manuscript provides an overview of childhood asthma, its prevalence, and the challenges faced by children with asthma in the school system. It also presents the main themes derived from the thematic analysis of the focus group discussion and discusses the implications for boarding schools and the facilities required to cater to students with asthma. Overall, the manuscript addresses an important topic and provides valuable insights. However, there are some areas that require attention and revision.

Major Comments:

1. Sample Size and Participant Selection: The manuscript does not provide a clear rationale for the selection of the 19 respondents from the initial 150 participants. It is important to provide a justification for the final sample size and explain any potential biases or limitations associated with the selection process.

2. Thematic Analysis: While the manuscript mentions that thematic analysis was conducted, it lacks details about the specific steps and analytical framework used. Providing more information on the process of thematic analysis and the specific themes that emerged would enhance the transparency and rigor of the study. Additionally, it would be helpful to include a brief discussion on the validity of the findings.

3. Generalizability: The manuscript should acknowledge the limitations regarding generalizability. The study was conducted within a specific social media group and included participants who are actively involved in asthma care. Therefore, the perspectives presented may not fully represent the broader population. It is recommended to discuss the limitations and suggest future research directions to include a more diverse range of participants.

4. Recommendations and Implications: The manuscript briefly mentions recommendations for boarding schools to create an inclusive and supportive environment for students with asthma. However, these recommendations should be further developed and expanded upon to provide actionable suggestions for policymakers, school administrators, and healthcare professionals. Consider discussing specific strategies for improving asthma management in boarding schools, such as the training of staff, collaboration with healthcare providers, and development of asthma-friendly environments.

Minor Comments:

1. Geographical Context: Consider providing additional contextual information on the Nigerian education system, including the prevalence of boarding schools and their characteristics. This would help readers better understand the specific context in which the study was conducted.

2. Limitations and Future Research: While this study provides valuable insights into the perceptions and opinions regarding boarding schools for children with asthma, it is important to acknowledge its limitations and identify areas for future research.

(a) Quantitative Assessment: This study adopted a qualitative design using focus group discussions. While qualitative methods are valuable in exploring in-depth perspectives and experiences, future research could complement these findings with quantitative assessments. Surveys or questionnaires could be administered to a larger sample size to quantify the prevalence of specific opinions and perceptions related to boarding schools for children with asthma. This would allow for statistical analysis and a more comprehensive understanding of the overall attitudes towards this topic.

(b) Longitudinal Studies: The current study provides a snapshot of participants' opinions at a specific point in time. Conducting longitudinal studies would enable the examination of changes in attitudes and perceptions over time. This would provide valuable insights into the dynamics of opinions regarding boarding schools for children with asthma and allow for a deeper understanding of the factors influencing these perspectives.

(c) Comparative Analysis: To gain a better understanding of the suitability of boarding schools for children with asthma, future research could compare the experiences and outcomes of children with asthma attending boarding schools versus day schools. This comparative analysis would shed light on the potential advantages and disadvantages of each educational setting and help inform policymakers, school administrators, and parents/guardians in making well-informed decisions.

(d) Intervention Studies: Building upon the findings of this study, future research could focus on developing and implementing interventions aimed at improving the management and support systems for children with asthma in boarding schools. Evaluating the effectiveness of interventions, such as asthma education programs for teachers and students, improved medical facilities and resources, and strategies to reduce stigma and promote inclusivity, would provide evidence-based recommendations for creating asthma-friendly environments in boarding schools.

Addressing these limitations and conducting further research in these areas would contribute to a more comprehensive understanding of the appropriateness of boarding schools for children with asthma and facilitate the development of evidence-based strategies to support their well-being and academic success.

Reviewer #2: The manuscript is very rich in vital scientific information that is of immense benefit to public and school health, which has been a neglected tropical issue. However, my major concern is the fact that out of 150 participants only 19 respondents' responses were available and used for discussions and drawing conclusions. The question now is, are these 19 responses sufficient to make the conclusion seen in this manuscript?

Secondly, why was sample size not determined for the study? This would have ensured that more responses and statistically significant population responses for concrete conclusion is gotten.

Thirdly, you stated that majority of respondents were is support of allowing children with asthma go to boarding school. What's your definition of majority? It is better you state the majority in numerical terms.

The use of social media and WhatsApp differently is a mere repetition.

The method(s) used were not properly cited and where modifications exist were not spelt out.

Many statements made in the discussion were not related to corroborate the work of others, it was mere hanging statement.

Some of the statements were also just made without linkages.

If the research was funded, the sponsors need to be acknowledged.

6. PLOS authors have the option to publish the peer review history of their article (what does this mean?). If published, this will include your full peer review and any attached files.

Reviewer #1: **Yes: **Oluwafemi Balogun, MD, MPH

Reviewer #2: No

---

## [Author Response · Author response to Decision Letter 0]

16 Oct 2023

RESPONSES TO THE REVIEWERS

PONE-D-23-13651: Are boarding secondary schools suitable for students with asthma? A focus group discussion

RESPONSES TO REVIEWER 1

MAJOR COMMENTS

1. Sample Size and Participant Selection: The manuscript does not provide a clear rationale for the selection of the 19 respondents from the initial 150 participants. It is important to provide a justification for the final sample size and explain any potential biases or limitations associated with the selection process.

See page 8: Sample size and selection

Sample size was not calculated. All registered members of the AACG WhatsApp® platform were eligible for participation. Convenience sampling technique, within a specified timeline, was employed. Comments were received from 19 participants who provided responses within the allotted timeframe for data collection.

See page 33: Limitations of the study

The study was conducted within the confines of a closed social media group, the Asthma Awareness and Care Group (AACG), which is predominantly made up of healthcare professionals and enthusiasts involved in asthma care. These participants may possess a higher level of knowledge and expertise regarding asthma management than the general population and may not fully represent a general perspective on the subject matter. Furthermore, the expertise and familiarity with the subject matter might have influenced their responses and recommendations. In addition, the study employed convenience sampling without calculating sample size. As a result, the sample size may not be sufficiently large or diverse to capture the wide range of opinions and experiences related to boarding schools and asthma management in Nigeria. Future studies should consider a more diverse range of participants.

Authors’ comments: Sampling of participants for the study and the limitations associated with the selection process are included in the manuscript.

2. Thematic Analysis: While the manuscript mentions that thematic analysis was conducted, it lacks details about the specific steps and analytical framework used. Providing more information on the process of thematic analysis and the specific themes that emerged would enhance the transparency and rigor of the study. Additionally, it would be helpful to include a brief discussion on the validity of the findings.

See Page 8: Methods; Data analysis section

For the identification of a thematic framework, nine researchers independently acquainted themselves with the transcripts, acquiring a general feel for the data and noting emergent ideas on the margins. The researchers identified key themes in the transcript. This was similar to a study conducted on challenges and expectations of the Mental Capacity Act of 2005 where an in-depth qualitative methodology was adopted. Three a priori questions formed the thematic framework, based on the codes and sub-themes that were subsequently identified [19]. The thematic framework included three central themes: 1) appropriateness of boarding school for children with asthma; 2) facilities necessary for boarding schools to cater for children with asthma; 3) outright rejection of children with asthma by boarding schools.

For the “indexing” stage, the nine researchers were assigned different parts of the transcripts, which they iteratively read to identify ideas and assign codes to portions of text. This generated a coding framework. This process was repeated until no new codes were generated. Discrepancies between the nine researchers' codes were resolved through discussion and a unified coding framework was obtained [20]. These codes were initially identified under the three broad themes in the thematic framework, after which similar codes were clustered to form sub-themes. 

In the “charting” step, these themes, subthemes and codes were presented in a chart and discussed at team meetings, enabling the re-ordering of the subthemes as necessary. This made it possible to extract relevant details from the data, including elements that may have been missed if a purely a priori approach was employed [21]. 

In the “mapping and interpretation” stages, the themes and subthemes were tabulated alongside exemplar quotes from participants. This process produced a 'tree structure' featuring interconnected themes, subthemes, and supporting quotes. Finally, these identified ideas were interpreted to answer the research questions and identify areas for public health action.

Authors’ comments: Modifications have been made to this section, in line with the Reviewer’s comments.

3. Generalizability: The manuscript should acknowledge the limitations regarding generalizability. The study was conducted within a specific social media group and included participants who are actively involved in asthma care. Therefore, the perspectives presented may not fully represent the broader population. It is recommended to discuss the limitations and suggest future research directions to include a more diverse range of participants.

See page 33: Limitations of the study

The study was conducted within the confines of a closed social media group, the Asthma Awareness and Care Group (AACG), which is predominantly made up of healthcare professionals and enthusiasts involved in asthma care. These participants may possess a higher level of knowledge and expertise regarding asthma management than the general population and may not fully represent a general perspective on the subject matter. Furthermore, the expertise and familiarity with the subject matter might have influenced their responses and recommendations. In addition, the study employed convenience sampling without calculating sample size. As a result, the sample size may not be sufficiently large or diverse to capture the wide range of opinions and experiences related to boarding schools and asthma management in Nigeria. Future studies should consider a more diverse range of participants.

Authors’ comments: Modifications have been made to highlight the limitations regarding generalizability of the findings of the study. A statement, on the need for future studies to include a more diverse range of participants, has been added.

4. Recommendations and Implications: The manuscript briefly mentions recommendations for boarding schools to create an inclusive and supportive environment for students with asthma. However, these recommendations should be further developed and expanded upon to provide actionable suggestions for policymakers, school administrators, and healthcare professionals. Consider discussing specific strategies for improving asthma management in boarding schools, such as the training of staff, collaboration with healthcare providers, and development of asthma-friendly environments.

See Page 32, 33

This study is an attempt to evoke thought process on whether children with asthma should be in boarding schools. It suggests that the suitability of boarding schools for children with asthma is dependent on many factors. Boarding schools should create an inclusive and supportive environment for students with asthma.

Boarding school administrators should ensure that they have adequate facilities to manage patients with asthma. This includes a functional sick bay and health professionals who would be available whenever needed. The sick bay should have medicines for the management of acute and chronic asthma, as well as asthma devices such as peak flow meters and nebulisers. 

The boarding school should have links with registered hospitals that can handle referrals. There should be collaborative efforts amongst the teachers, students, parents and school administrators to ensure students with asthma have good quality of life while at school. 

Asthma education programmes should be conducted for all active players (students, parents, teachers, school administrators). There should be trainings and re-trainings for staff and active players who can serve as first-responders if there is an asthma attack.

The boarding school should have asthma-friendly environments. All possible triggers for asthma should be eliminated.

Policy makers and those who approve schools to take boarding students should ensure that the facilities and environments are suitable for patients with asthma. Lists of items that need to be put in place should be provided to school administrators. Policy makers should form a think tank using this study as baseline with a view to providing policy framework that will guide the educational sector to make important changes to the boarding schools that would host students with asthma.

Future studies should consider quantitative assessments. Questionnaires could be administered to a larger sample size to quantify the prevalence of specific opinions and perceptions related to boarding schools for children with asthma. Longitudinal studies could also be conducted to provide valuable insights into the dynamics of opinions regarding boarding schools for children with asthma and allow for a deeper understanding of the factors influencing these perspectives. Mixed-method analysis could be done to better understand the experiences and health outcomes of children with asthma attending boarding schools in comparison to those attending day schools. 

Building upon the findings of this study, future research could focus on developing and implementing interventions aimed at improving the management and support systems for children with asthma who are in boarding schools. Evaluating the effectiveness of interventions, such as asthma education programmes for teachers and students, improved medical facilities and resources, and strategies to reduce stigma and promote inclusivity could provide evidence-based recommendations for creating asthma-friendly environments in boarding schools.

Author’s comments: Modifications have been made, in line with the suggestions of the Reviewer.

MINOR COMMENTS

1. Geographical Context: Consider providing additional contextual information on the Nigerian education system, including the prevalence of boarding schools and their characteristics. This would help readers better understand the specific context in which the study was conducted.

See Page 5: Introduction section

Nigeria currently follows the Universal Basic Education (9-3-4 system of education), a transition from the 6-3-3-4 system [7]. The 9-3-4 system entails nine years of basic education, three years of senior secondary education, and four years of tertiary education. The curriculum for the 9-3-4 system of education was designed to meet the Millenium Development Goals (MDGs) by 2020 [7]. Students can be in boarding schools from the seventh to ninth year of basic education and during the three years of senior secondary education. There is an option of having children going from home (day students) or living and studying in the school during the school term (boarders) [8]. 

In Nigeria, 5 of every 100 adolescents in Nigeria who are school children, especially those in secondary schools, are in boarding schools [9]. Many parents, lean towards the option of boarding schools as it helps children build independence, limits access to distracting technology, and provides an established learning community where they can easily form study groups [10].

Author’s comments: Modifications have been made, in line with the Reviewer’s comments.

2. Limitations and Future Research: While this study provides valuable insights into the perceptions and opinions regarding boarding schools for children with asthma, it is important to acknowledge its limitations and identify areas for future research.

(a) Quantitative Assessment: This study adopted a qualitative design using focus group discussions. While qualitative methods are valuable in exploring in-depth perspectives and experiences, future research could complement these findings with quantitative assessments. Surveys or questionnaires could be administered to a larger sample size to quantify the prevalence of specific opinions and perceptions related to boarding schools for children with asthma. This would allow for statistical analysis and a more comprehensive understanding of the overall attitudes towards this topic.

(b) Longitudinal Studies: The current study provides a snapshot of participants' opinions at a specific point in time. Conducting longitudinal studies would enable the examination of changes in attitudes and perceptions over time. This would provide valuable insights into the dynamics of opinions regarding boarding schools for children with asthma and allow for a deeper understanding of the factors influencing these perspectives.

(c) Comparative Analysis: To gain a better understanding of the suitability of boarding schools for children with asthma, future research could compare the experiences and outcomes of children with asthma attending boarding schools versus day schools. This comparative analysis would shed light on the potential advantages and disadvantages of each educational setting and help inform policymakers, school administrators, and parents/guardians in making well-informed decisions.

(d) Intervention Studies: Building upon the findings of this study, future research could focus on developing and implementing interventions aimed at improving the management and support systems for children with asthma in boarding schools. Evaluating the effectiveness of interventions, such as asthma education programs for teachers and students, improved medical facilities and resources, and strategies to reduce stigma and promote inclusivity, would provide evidence-based recommendations for creating asthma-friendly environments in boarding schools.

See Page 33:

Future studies should consider quantitative assessments. Questionnaires could be administered to a larger sample size to quantify the prevalence of specific opinions and perceptions related to boarding schools for children with asthma. Longitudinal studies could also be conducted to provide valuable insights into the dynamics of opinions regarding boarding schools for children with asthma and allow for a deeper understanding of the factors influencing these perspectives. Mixed-method analysis could be done to better understand the experiences and health outcomes of children with asthma attending boarding schools in comparison to those attending day schools. 

Building upon the findings of this study, future research could focus on developing and implementing interventions aimed at improving the management and support systems for children with asthma who are in boarding schools. Evaluating the effectiveness of interventions, such as asthma education programmes for teachers and students, improved medical facilities and resources, and strategies to reduce stigma and promote inclusivity could provide evidence-based recommendations for creating asthma-friendly environments in boarding schools.

Author’s comments: Modifications have been made, in line with the Reviewer’s comments.

RESPONSES TO REVIEWER 2

1. The manuscript is very rich in vital scientific information that is of immense benefit to public and school health, which has been a neglected tropical issue. However, my major concern is the fact that out of 150 participants only 19 respondents' responses were available and used for discussions and drawing conclusions. The question now is, are these 19 responses sufficient to make the conclusion seen in this manuscript?

See Page 33 (Conclusion):

This study, in an attempt to address the issue of the appropriateness of boarding school for children with asthma, has identified children’s age, autonomy, asthma management status, and the school’s readiness, as important considerations for the safe attendance of children with asthma at boarding schools.

Author’s comments: The small sample size is included as one of the limitations of the study. The study provides a baseline for future studies. It is an attempt to generate some insights and thought-provoking discussions on the subject matter.

2. Secondly, why was sample size not determined for the study? This would have ensured that more responses and statistically significant population responses for concrete conclusion is gotten.

See Page 8 (Sample size and selection):

Sample size was not calculated. All regist

---

## [Decision Letter · Decision Letter 1]

24 Mar 2024

PONE-D-23-13651R1Are boarding secondary schools suitable for students with asthma? A focus group discussionPLOS ONE

Dear Dr. Amorha,

Thank you for submitting your manuscript to PLOS ONE. After careful consideration, we feel that it has merit but does not fully meet PLOS ONE’s publication criteria as it currently stands. Therefore, we invite you to submit a revised version of the manuscript that addresses the points raised during the review process.

We look forward to receiving your revised manuscript.

Kind regards,

Nabeel Al-Yateem, PhD

Academic Editor

PLOS ONE

Journal Requirements:

Reviewers' comments:

Reviewer's Responses to Questions

**Comments to the Author**

1. If the authors have adequately addressed your comments raised in a previous round of review and you feel that this manuscript is now acceptable for publication, you may indicate that here to bypass the “Comments to the Author” section, enter your conflict of interest statement in the “Confidential to Editor” section, and submit your "Accept" recommendation.

Reviewer #1: All comments have been addressed

Reviewer #3: All comments have been addressed

2. Is the manuscript technically sound, and do the data support the conclusions?

Reviewer #1: Yes

Reviewer #3: Partly

3. Has the statistical analysis been performed appropriately and rigorously? 

Reviewer #1: Yes

Reviewer #3: N/A

4. Have the authors made all data underlying the findings in their manuscript fully available?

Reviewer #1: (No Response)

Reviewer #3: Yes

5. Is the manuscript presented in an intelligible fashion and written in standard English?

Reviewer #1: (No Response)

Reviewer #3: Yes

6. Review Comments to the Author

Reviewer #1: (No Response)

Reviewer #3: An interesting study. The authors should consider the following comments mainly suggestions:

• In the title, possibly add the type of FGD or participants.

• L42 – indicate “parents or caregivers”, for the rest of the document, the authors should indicate that reference to parents includes caregivers.

• L126-127, 156/7 - the authors note that demographic data were collected – any more apart from age and perhaps profession? It could have been useful in analysis.

• I suggest that participants refer to the 19 not `150; 150 were contacted and 19 responded.

• L48 – there is no evidence that all the 150 members participated.

• L150-155 – This looks more like a vignette.

• I wonder whether it is appropriate to quantify such small numbers (61.5%) when it is a purely qualitative study.

7. PLOS authors have the option to publish the peer review history of their article (what does this mean?). If published, this will include your full peer review and any attached files.

Reviewer #1: **Yes: **Oluwafemi Balogun, MD, MPH

Reviewer #3: No

---

## [Author Response · Author response to Decision Letter 1]

25 Mar 2024

RESPONSES TO THE REVIEWERS

PONE-D-23-13651R1: Are boarding secondary schools suitable for students with asthma? A focus group discussion

COMMENTS AND RESPONSES 

1. In the study title, possibly add the type of FGD or participants.

See page 1, Lines 1 - 3: Study title

Are boarding secondary schools suitable for students with asthma? An asynchronous online focus group discussion among members of an asthma awareness group

Authors’ comments: The type of focus Group Discussion and participants have been included in the title.

 2. Indicate “parents or caregivers”, for the rest of the document, the authors should indicate that reference to parents includes caregivers.

See page 3, Lines 43, 44

See page 4, Line 90

See page 6, Line 97

See page 29, Theme 1, Second Paragraph

Authors’ comments: Parents or caregivers have been indicated all through the document.

3. The authors note that demographic data were collected – any more apart from age and perhaps profession? It could have been useful in analysis.

Authors’ comments: The only demographic data collected were age, gender, and profession. The analysis was focused on generating themes from the different responses. The authors did not deem it necessary to request for more demographic data as a test of association between demographic data and themes was not included as a study objective. 

If it is highly-important, it can be included as a limitation of the study.

We have included the Participant Number (e.g., P1, P2 etc.) in the Tables.

4. I suggest that participants refer to the 19 not `150; 150 were contacted and 19 responded. L48 – there is no evidence that all the 150 members participated.

See page 3, Line 49: Abstract

The group comprised 150 registered members.

See page 3, Line 52: Abstract

Results: Out of the 150 eligible members, there were responses from only 19 participants.

Authors’ comments: Corrections have been made in the Abstract section and Main Text, as suggested, to eliminate ambiguities. Although the 150 registered members in the WhatsApp group received the messages (there is evidence that it was delivered to al)l, only 19 members responded with comments. 

5. L150-155 – This looks more like a vignette.

See page 3, Line 50: Abstract

… based on a vignette comprising three questions.

Page 7, Line 121: Study design

The vignette was posted in the Group, without previous publicity.

See page 8, Line 151: Data collection

This vignette was posted on the AACG WhatsApp® platform:

Authors’ comments: Corrections have been made in the Abstract section and Main Text. 

6. I wonder whether it is appropriate to quantify such small numbers (61.5%) when it is a purely qualitative study.

See Page 3, Line 62

See Page 29, Theme 1, Second Paragraph

Authors’ comments: The percentage was not previously stated. It was included based on the request by one of the Reviewers. It provides an idea about the proportion of respondents who supported that statement. Maybe the inclusion or exclusion of the percentage figure should be left for the discretion of the Editor.

General Response

In addition, typographical and grammatical errors have been corrected in this revision. All edits are in red fonts.

---

## [Editor Report · Decision Letter 2]

7 May 2024

Are boarding secondary schools suitable for students with asthma? An asynchronous online focus group discussion among members of an asthma awareness group

PONE-D-23-13651R2

Dear Dr. Amorha,

We’re pleased to inform you that your manuscript has been judged scientifically suitable for publication and will be formally accepted for publication once it meets all outstanding technical requirements.

Kind regards,

Nabeel Al-Yateem, PhD

Academic Editor

PLOS ONE
---

## [Editor Report · Acceptance letter]

14 Jun 2024

PONE-D-23-13651R2 

PLOS ONE

Dear Dr. Amorha, 

I'm pleased to inform you that your manuscript has been deemed suitable for publication in PLOS ONE. Congratulations! Your manuscript is now being handed over to our production team.

Kind regards, 

on behalf of

Dr. Nabeel Al-Yateem 

Academic Editor

PLOS ONE